# Weldability Evaluation of Alloy 718 Investment Castings with Different Si Contents and Thermal Stories and Hot Cracking Mechanism in Their Laser Beam Welds

Pedro Álvarez [1,*], Alberto Cobos [1], Lexuri Vázquez [1], Noelia Ruiz [2], Pedro Pablo Rodríguez [2], Ana Magaña [3], Andrea Niklas [3] and Fernando Santos [3]

[1]  LORTEK Technological Centre, Basque Research and Technology Alliance (BRTA), 20240 Ordizia, Spain; acobos@lortek.es (A.C.); lvazquez@lortek.es (L.V.)
[2]  EIPC RESEARCH CENTER, AIE, Torrekua 3, 20600 Eibar, Spain; nruiz@eipc.es (N.R.); prodriguez@eipc.es (P.P.R.)
[3]  Fundación AZTERLAN, Basque Research and Technology Alliance (BRTA), Aliendalde Auzunea 6, 48200 Durango, Spain; amagana@azterlan.es (A.M.); aniklas@azterlan.es (A.N.); fsantos@azterlan.es (F.S.)
*  Correspondence: palvarez@lortek.es; Tel.: +34-943-88-23-03

**Abstract:** In this work, weldability and hot cracking susceptibility of five alloy 718 investment castings in laser beam welding (LBW) were investigated. Influence of chemical composition, with varying Si contents from 0.05 to 0.17 wt %, solidification rate, and pre-weld heat treatment were studied by carrying out three different weldability tests, i.e., hot ductility, Varestraint, and bead-on-plate tests, after hot isostatic pressing (HIP) and solution annealing treatment. Onset of hot ductility drop was directly related to the presence of residual Laves phase, whereas the hot ductility recovery behaviour was connected to the Si content and γ grain size. LBW Varestraint tests gave rise to enhanced fusion zone (FZ) cracking with much more reduced heat-affected zone (HAZ) cracking that was mostly independent of Si content and residual Laves phase. Microstructural characterisation of bead-on-plate welding samples showed that HAZ cracking susceptibility was closely related to welding morphology. Multiple HAZ cracks were detected in nail or mushroom welding shapes, typical in keyhole mode LBW, irrespective of the chemical composition and thermal story of castings. In all LBW welds, Laves phase with a composition similar to the eutectic of the pseudo-binary equilibrium diagram of alloy 718 was formed in the FZ. The composition of this regenerated Laves phase matched with the continuous Laves phase film observed along HAZ cracks. This was strong evidence of backfilling mechanism, which is described as wetting and infiltration of terminal liquid along γ grain boundaries of parent material. The current results suggest that this cracking mechanism was activated in three-point intersections resulting from perpendicular crossing of columnar grain boundaries with fusion line and was enhanced by nail or mushroom weld shapes and narrow and columnar γ grain characteristics of castings. Neither Varestraint nor hot ductility weldability tests can reproduce this particular cracking mechanism.

**Keywords:** investment casting; alloy 718; hot cracking mechanism; Varestraint test; laser beam welding

## 1. Introduction

Alloy 718 was developed almost 60 years ago [1] and it has been the most widely used Ni-based superalloy to date. Being a precipitation-strengthened superalloy, it has been broadly used for the manufacturing of both land-based energy and aircraft turbine components, showing an outstanding performance at working temperatures up to 700 °C under high structural loading and corrosive conditions [2].

While other precipitation-hardened Ni superalloys have relatively high amounts of gamma prime (γ′) former elements, i.e., Al and Ti, alloy 718 is based on the addition of Nb, which forms metastable gamma double-prime (γ″) precipitates of Ni$_3$Nb. The precipitation

kinetics of $\gamma''$ is slower compared to $\gamma'$ (Ni$_3$Al, Ni$_3$Ti and Ni$_3$(Ti, Al)), which contributes to improve castability, hot working, and weldability. The improvement is basically due to the fact that alloy 718 remains in a softer state during these manufacturing processes, avoiding the build-up of internal stresses. In terms of weldability, the sluggish precipitation kinetics of alloy 718 minimises strain age cracking (SAC) after welding and during post-welding heat treatment [2].

Alloy 718 investment castings are usually melted and poured inside vacuum furnaces and subsequently heat treated by hot isostatic pressing (HIP) to ensure highest performance. Intermetallic phases such as NbC and Nb-rich Laves phase can be found in alloy 718 castings due to segregation of chemical elements during slow solidification [3,4]. These secondary phases solidify at low temperatures (e.g., $\gamma$/Laves eutectic at temperatures down to 1180 °C) and they are usually concentrated along grain boundaries. Melting of Laves phases that are formed during the terminal solidification has been identified as the origin of the higher cracking susceptibility of castings during welding in comparison with wrought alloy 718 [4,5]. The reason for this is that Laves phase is readily melted upon heating in contrast with constitutional liquation of NbC that requires a dissolution reaction to form a liquid [2,6]. The latter is the predominant liquation mechanism identified in welds of wrought parts [2,6,7].

Incipient melting of Laves phase gives rise to a liquid which is distributed along grain boundaries, drastically reducing the strength of the material and its capability to withstand stresses. Therefore, sophisticated thermal treatments have been developed with the aim of reducing the amount of Laves phases and consequently heat-affected zone (HAZ) liquation cracking susceptibility of alloy 718 castings [4,8,9]. The aim of these treatments is to solubilise deleterious Laves phase and reduce compositional gradients in the as-cast microstructure. In fact, the concentration of several residual elements such as B, P, and S in grain boundaries can promote HAZ liquation cracking to a higher degree by decreasing even more the initial melting temperature and modifying the wetting characteristics of the intergranular liquid [4,10–12].

Formation of $\gamma$/Laves eutectic can also cause fusion zone (FZ) cracking in alloy 718 welds. This eutectic solidifies at a much lower temperature than the bulk matrix and wi-dens the solidification temperature range [2,4]. Wider solidification temperature range is directly associated with higher hot cracking susceptibility due to a longer coexistence of solid and liquid phases.

In order to study hot cracking susceptibility of Ni superalloys, researchers have defined and implemented different weldability assessment trials [2]. They are usually classified into three categories.

Self-restrained or representative tests use the inherent strain of the welding to induce cracking. They try to reproduce real joint configuration and residual stress levels. The drawback of this type of test is that it does not give quantitative values of cracking susceptibility and the result only indicates if the weld cracks pop up or not. Sometimes circular welding paths are applied to induce higher residual stresses.

In simulative tests, either a tension or bending deformation is externally applied du-ring welding. The Varestraint test is probably the best known test in this category, entai-ling the application of bending deformation along the longitudinal direction of the weld. The development of the Varestraint testing method from its origin was thoroughly reviewed by Andersson et al. [13]. Deformation enhances hot cracking and its extension, i.e., number and length of cracks, which can be determined at different strain levels. In this way, cracking susceptibility of different materials can be compared [14,15]. However, in real welding applications, residual stresses are predominant in contrast with plastic strain.

Finally, both strength and ductility are directly measured at high temperatures in hot ductility tests. In these tests, the temperature at which the material loses complete its strength (nil strength temperature (NST)) is determined by heating up testing samples under a constant tensile load. Additionally, nil ductility temperature (NDT), corresponding to the peak heating temperature at which the area reduction of the broken surface

is 0%, and the ductility recovery temperature (DRT), at which 5% of area reduction is reco-vered after cooling down from a temperature close to NST, are computed. NDT and DRT are determined from on-cooling curves and the strain is only applied when the testing temperature has been reached. Hot ductility behaviour is linked to hot cracking susceptibility since cracks are generated when the material cannot accommodate stresses and strains induced during welding [2,9].

In a recent paper [16], the current authors compared the hot cracking susceptibility of wrought and investment casting alloy 718 by Varestraint test while applying pulsed and continuous tungsten inert gas (TIG) welding and LBW. It was concluded that hot cracking was enhanced in LBW samples due to extended centre line fusion zone (FZ) cracking showing a fishbone-like cracking pattern. Minor influence of pulsation mode and grain size was observed and, in fact, casting samples with grain sizes 30 times coarser showed slightly better performance than wrought material. It must be noted that Laves phases were not observed in investment casting samples and only some traces of needle-like delta (δ) phase and Mo sulphide were detected in base material.

Pulsed current can refine solidification microstructure and reduce the amount of Laves phase and Nb segregations in TIG welds according to [17,18]. Moreover, Bai et al. [19] recently investigated the potential benefits of combining high-frequency micro-vibration and LBW. Under particular vibration frequencies, the length of the liquation cracks in HAZ was reduced, but not completely avoided. The authors performed bead-on-plate tests and obtained weld cross sections with nail or mushroom shape. LBW has also been studied in alloy 718 parts by other researchers [20–25]. These investigations targeted the influence of LBW parameters and energy input on porosity, microstructure, and mechanical properties of these welds. A deep analysis of HAZ cracking susceptibility of this alloy during LBW was performed in particular by [20,22]. Impact of weld shape morphology, grain size, pre-weld heat treatment, and boron segregation were investigated.

In this work, weldability and hot cracking susceptibility in laser beam welds of five alloy 718 investment castings were investigated. Influence of chemical composition, Si content, solidification rate, and pre-weld heat treatment was studied by carrying out three different weldability tests. Cracking behaviour was compared, and results were completed with the detailed microstructural analysis after welding tests. Additionally, influence of pre-weld heat treatment on microstructure of parent material is discussed. An analysis about the correlation of weldability assessment test results, i.e., Varestraint and hot ductility, with cracking trend observed in real bead-on-plate LBW trials was performed. Fundamentals of the mechanism that triggers HAZ cracking in bead-on-plate test are explained.

## 2. Materials and Methods

### 2.1. Investment Casting and Chemical Composition of Alloy 718 Casting Heats

Investment casting moulds were manufactured, incorporating 20 test samples as flat plates of 150 × 50 × 10 mm in each mould. Moulds were covered with a ceramic shell composed by three primary layers and five backup layers, dewaxed, preheated, poured, and later demoulded, as described in Figure 1.

Casting samples were cast at industrial facilities under vacuum conditions. The mould preheating temperature was always 1150 °C, whereas the molten material was poured at 1450 °C.

The chemical composition of the different castings is shown in Table 1. The content of each alloying element was determined in cast samples using the following analytic techniques. C and S contents were measured by combustion and infrared absorption; N by inert gas fusion and thermal conductivity; O by inert gas fusion and infrared absorption; and finally Si, Mn, P, Fe, Cr, Mo, Ti, Al, and Nb contents were determined by spark atomic emission spectrometry. Co and B contents were not measured in casted samples, but they were taken from the chemical composition of the ingots employed as raw material for castings averaging 0.11 wt % and 0.002 wt %, respectively.

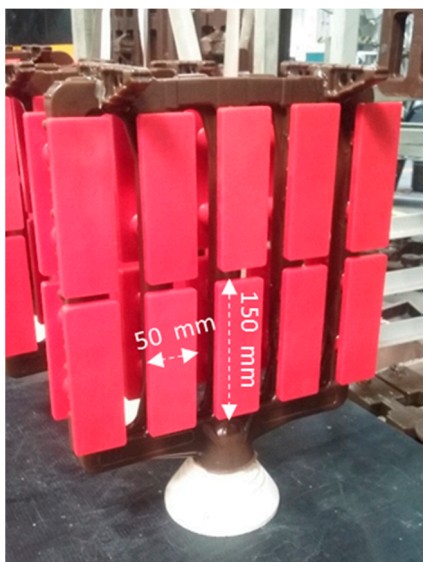

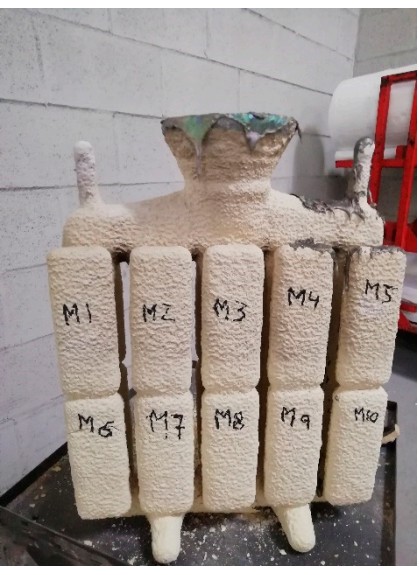

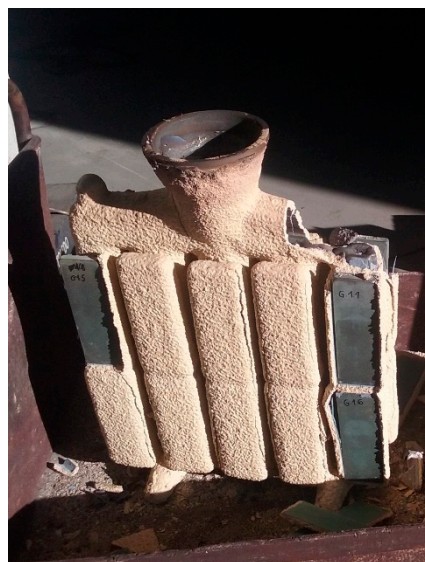

(**a**) Mould in wax  (**b**) Mould with shell  (**c**) Mould without shell

**Figure 1.** Investment casting moulds in different manufacturing steps.

**Table 1.** Chemical composition of alloy 718 casting heats in weight percentage.

| Ref. | Ni | C | Si | Mn | P | S | Fe | Cr | Mo | Ti | Al | Nb + Ta |
|------|------|-------|-------|--------|--------|--------|------|------|------|------|------|---------|
| O | 51.9 | 0.047 | 0.051 | <0.050 | <0.010 | <0.005 | 21.1 | 17.8 | 3.02 | 0.89 | 0.47 | 4.75 |
| E | 52.1 | 0.049 | 0.11 | 0.037 | <0.010 | <0.005 | 20.4 | 17.6 | 2.91 | 0.98 | 0.59 | 4.92 |
| P | 51.7 | 0.038 | 0.17 | <0.050 | <0.010 | <0.005 | 21.1 | 17.7 | 3.02 | 0.89 | 0.46 | 4.85 |
| N/NP | 52.5 | 0.058 | 0.12 | 0.038 | <0.010 | <0.005 | 20.3 | 17.7 | 2.88 | 0.77 | 0.47 | 4.88 |

Note that mould O (low Si content) was manufactured by using high purity ingots as raw material. In mould P (high Si content), Si was intentionally added during vacuum melting, and chemically adjusted ingots (28 Kg in total) were manufactured in a first melting step. Mould E corresponded to conventional chemical composition and casting process, whereas the cooling rate during solidification was reduced in moulds N and NP by incorporating 1 ceramic blanket over cast parts. The cooling rates between 800 and 500 °C were determined and they were 0.52 °C/s and 1.65 °C/s for the moulds with blanket (ref. N/NP) and without blanket (ref. O/E/P), respectively.

Once parts were shot blasted, they were submitted to a heat treatment process, according to GKN V.AC:9922 standard and comprising HIP and solution annealing thermal cycles. An additional solubilisation pre-HIP treatment consisting of solubilisation at 1052 °C for 2 h followed by air cooling was applied to slowly cooled mould NP. The goal of this treatment was to dissolve Laves phases before HIP treatment.

### 2.2. Weldability Assessment Trials

Three different weldability assessment trials were carried out with the 5 investigated alloy 718 casting heats after completing heat treatments described in the previous section, thus, in the solution annealing state. LBW Varestraint tests were carried out in a testing device (Figure 2) that was fully designed and manufactured at LORTEK [15,16]. Performance of the test bench complied with general requirements of ISO/TR 17643-1 "Destructive tests on welds in metallic materials—Hot cracking tests for weldments—Arc welding processes—Part 3: Externally loaded tests" [26]. LBW Varestraint tests were performed on 3.2 mm thickness samples that were electric discharge machined (EDM) from 10 mm thickness casting plates. External surfaces were milled before testing. LBW was applied on the surface of testing samples without adding any filler metal by TRUDISK 6002 disk laser from TRUMPF company, Ditzingen, Germany. The laser beam was guided through

400 µm diameter fibre to a TRUMPF BEO D70 laser welding head (200 mm focus length and 200 mm collimation length). LBW Varestraint tests were completed at 0.5 m/min welding speed, 2300 W continuous mode power, and 0.8 mm diameter spot size. Tests were performed in a closed chamber filled by argon gas to avoid surface oxidation and provide good shielding conditions.

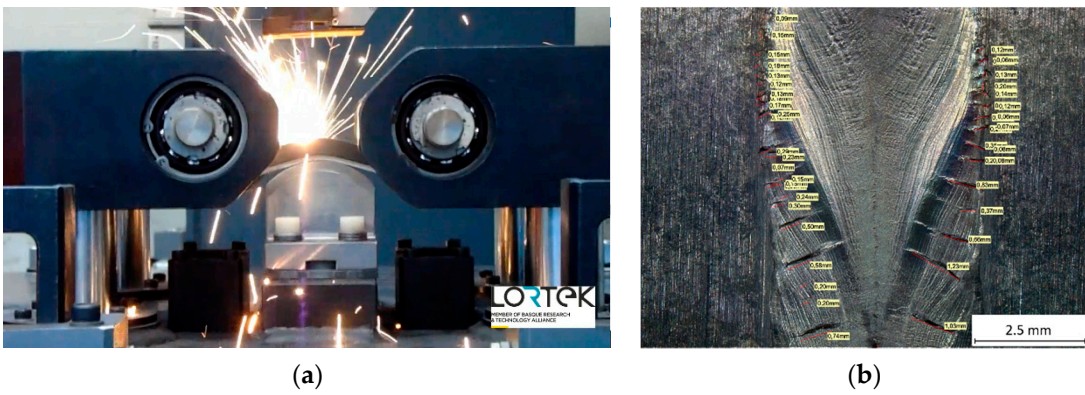

(**a**)          (**b**)

**Figure 2.** (**a**) LBW Varestraint testing device and (**b**) measurement of total crack lengths (TCL). Labels of individual cracks in fusion zone (FZ) and heat-affected zone (HAZ) are displayed.

Different augmented strains ($\varepsilon$) were applied during LBW by bending the samples along their length at 150 mm/s stroke rate and employing several interchangeable die blocks with radii varying from 20 to 320 mm. The induced augmented strains which are calculated by the following equation were in the range from 0.5 to 8%.

$$\varepsilon = \frac{t}{2 \cdot R} \times 100, \tag{1}$$

where $\varepsilon$ is the resulting augmented strain as a percentage, $t$ is the thickness of the sample in millimetres, and $R$ is the radius of the die block in millimetres. Two expendable support plates of 304 stainless steel were positioned in both sides of the testing samples to avoid kinking. Both FZ and HAZ cracking susceptibility were studied by determining total crack lengths (TCL) in these two zones of the welds [2,13,14].

Note that in this case, cracks were measured on the surface of the testing samples using magnification lenses (up to 150×) and after cleaning the surface of the welds by soft manual polishing and oxalic acid electroetching to avoid reflections.

Additionally, hot ductility tests of 5 casting heats were carried out in Gleeble 3800D thermomechanical simulator (DYNAMIC SYSTEMS INC., Austin, USA) owned by West University in Sweden. Here, 6 mm (−0.025 mm, +0.01 mm) diameter cylindrical shape samples were finely turned from 10 mm thickness as-cast plates. Hot ductility testing setup and guidelines included in Gleeble Users Training 2010 handbook were applied. These are comparable to the testing specifications included in procedure B of [26], with minimum differences in samples length. NST temperature was only determined in mould E, concentrating the overall weldability assessment of 5 alloys on on-heating and on-cooling tests. A heating rate of 111 °C/s from room to testing temperatures was employed in on-heating tests, whereas samples were heated up to 1195 °C at the same heating rate and subsequently cooled down at 50 °C/s to each testing temperature in on-cooling trials. Temperature profile was recorded with K-type thermocouple welded to the surface of testing samples in the area between clamps. The samples were pulled to fracture at 55 mm/s stroke rate. Percentages of area reduction from initial 28.3 mm² (i.e., 6 mm diameter) were measured to determine ductility at the different testing temperatures.

Finally, self-restrained bead-on-plate LBW trials were carried out in casting plates. In these representative tests, the parent materials were remelted by scanning the surface with a laser beam with the same energy distribution, shielding conditions and parameters

employed in LBW Varestraint tests. In this case, trials were carried out in samples with less than 3 mm thickness (between 2.6 and 2.9 mm) and 9 mm thickness, the latter resulting from the surface grinding of casting samples. Cross-sections of bead-on-plate samples were metallographically characterised to detect cracks in FZ and HAZ.

*2.3. Metallographic Characterisation*

As-cast, heat-treated, and bead-on-plate welding samples were characterised by optical microscopy (OM) in a LEICA MEF4 microscope (LEICA MICROSYSTEMS GmbH, Wetzlar, Germany) and field emission scanning electron microscopy (FESEM) with a ZEISS Ultra Plus microscope (CARL ZEISS AG, Oberkochen, Germany). Energy-dispersive X-ray (EDX) spectroscopy analysis was conducted in the FESEM microscope to determine local chemical composition of precipitates and phases. Average chemical composition of the Laves phases was determined through EDX analysis of 5 phases for each casting heat. For the metallographic analysis, cross sections were prepared by grinding and polishing using standard procedures. The area percentage of carbides was determined by image analysis of 5 images obtained by OM at 100× (HAZ and BM) and 500x (FZ) using Leica application suite V4.2. The area percentage of Laves phases + carbides was determined through SEM at 500× in the heat-affected zone (HAZ) and in the base metal (BM), and at 2500× in the fusion zone (FZ). Finally, the Laves phase area percentage was obtained by subtracting the carbide area percentage obtained by OM from the area percentage of Laves phases + carbides obtained by SEM.

The microsegregation degree of alloying elements was evaluated by the segregation ratio (SR), which can be calculated using the following formula [27]:

$$SR = C_{i;IR}/C_{i;DC} \tag{2}$$

where $C_{i,IR}$ is the maximum concentration of element i in interdendritic region, and $C_{i,DC}$ is the minimum concentration of element i in dendrite core. For SR < 1, solute elements tend to segregate to dendrite core during solidification; when SR > 1, the alloying elements partition towards the interdendritic region. If SR values are close to 1, the corresponding elements do not favourably segregate to any region. The concentration of each element was determined by EDX analysing 8 points in 3 different regions; an example is shown in Figure 3.

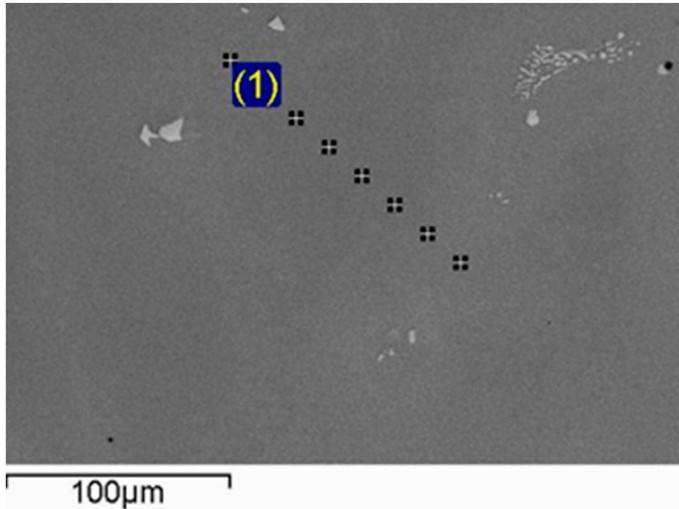

**Figure 3.** Location of EDX analysis points for determining the segregation ratio.

Grain size was revealed by etching with Kalling 2 reagent. Dimensions of at least 10 grains were analysed for each casting heat in cross-sections. The grain size was de-

termined from the horizontal and vertical mean intersection lengths between 2 grain boundaries of grains.

The solidification path and evolution of Laves phases were studied by Thermo-Calc software (using TCNI10 and MOBNI4 databases). Scheil simulation, considering back diffusion of alloying elements in the primary phase, was performed at a high cooling rate of 100 °C/s, similar to the expected welding cooling rates, for alloys with different Si contents (moulds O, E, and P).

## 3. Results

### 3.1. As-Cast Microstructure

The secondary phases observed in the five moulds or casting heats in as-cast condition are shown in Figure 4. The as-cast microstructure of the five moulds consisted of a γ matrix, Laves phases, Nb carbides (NbC), and smaller quantities of TiNb carbonitrides (TiNbCN). In the moulds with slow solidification rate, additional δ and γ″ phases were observed in the segregated interdendritic region. Laves phases and carbides show similar colour and also morphologies when analysed by SEM; thus, for identification it is necessary to perform an EDX analysis. Therefore, for the quantification of the Laves phase area percentage included in Table 2, first the area percentage of Laves phases + carbides was determined by SEM images. Second, the area fraction of carbides was evaluated by optical microscopy, which revealed in the unetched state only the presence of carbides. Finally, as it is also explained in Section 2.3, the Laves phase area percentage was obtained by subtracting the area percentage of carbides obtained by OM from the area percentage obtained by SEM (Laves + carbides). It can be observed that, as the Si content of the alloy increased from 0.051 wt % (mould O) to 0.17 wt % (mould P), the area percentage of Laves phases increased from 2.2% to 3.5% and the Si content in the Laves phases increased from 0.28 to 1.29 wt %.

High segregation of alloying elements to the interdendritic spaces, particularly for Nb and Ti but also Mo, was observed in the as-cast state of every mould. A slight depletion in Fe and Cr was observed in those interdendritic spaces (Table 3).

The characteristics of the γ grains are depicted in Table 4. It can be observed that the grains of moulds fabricated with normal solidification rate (moulds O, E, P) showed a highly columnar grain morphology and an aspect ratio between 3.0 and 3.2, while the slowly solidified moulds presented a coarser grain size and aspect ratio below 2.0. Mean width was measured as the mean horizontal distance between grain boundaries, whereas mean length corresponded to distance along sample thickness. Morphology and grain size of γ grains in as-cast samples is shown in Figure 5.

### 3.2. Microstructure after Heat Treatment and before Welding Trials

Heat treatment did not modify the as-cast grain size and morphology observed in Figure 5, however, segregation and amount of Laves phase were significantly reduced in comparison with as-cast condition as can be concluded by comparing Tables 3 and 5. The characteristics of the Laves phases after heat treatment (HIP+S) are shown in Table 2, together with the ones of as-cast samples. No Laves phases were detected in the low and standard Si-bearing alloys (moulds O and E), but small area percentages between 0.14 and 0.35% remained in high-Si alloy (mould P) and slowly solidified alloy with (mould NP) and without pre-HIP (mould N) heat treatments. It is also worthwhile mentioning that after heat treatment, the Laves phase composition was enriched in Mo and Si.

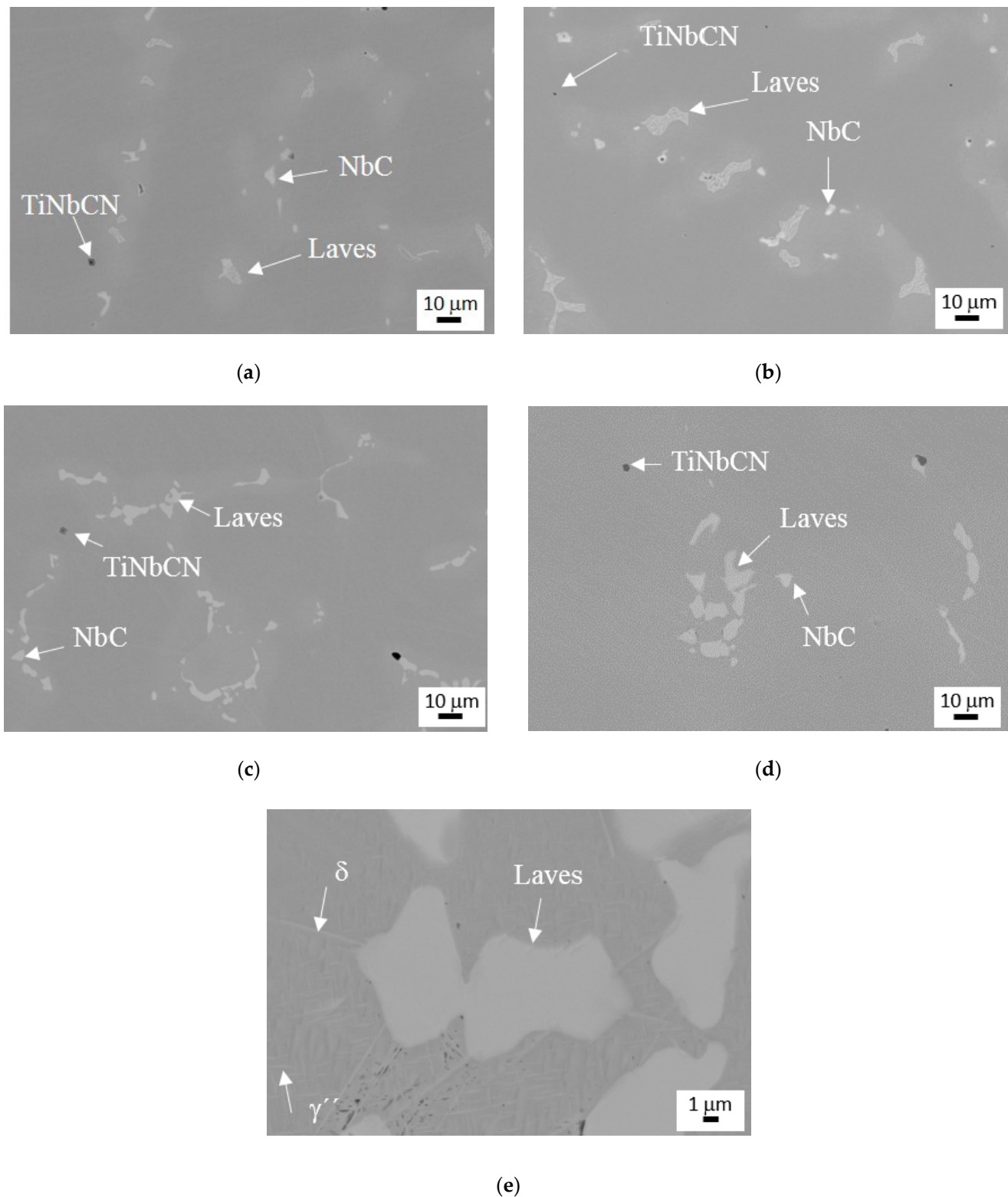

**Figure 4.** Secondary phases present in the microstructure of different moulds in the as-cast state: (**a**) O, (**b**) E, (**c**) P, and (**d**) NP; (**e**) NP: magnification of (**d**) showing the presence of Laves, δ phase, and γ″ in the interdendritic region.

**Table 2.** Area percentage and chemical composition of Laves phases (in wt %) as-cast and after heat treatment obtained by / EDX) analysis (* Laves phase was not detected).

| Mould | State | Area % Laves | Al | Si | Ti | Cr | Fe | Ni | Nb | Mo |
|---|---|---|---|---|---|---|---|---|---|---|
| O | As-cast | 2.20 | 0.13 ± 0.05 | 0.28 ± 0.06 | 0.91 ± 0.07 | 11.86 ± 0.05 | 11.98 ± 0.24 | 34.02 ± 0.64 | 33.90 ± 0.50 | 7.67 ± 0.50 |
|  | HIP + S * | 0 | - | - | - | - | - | - | - | - |
| E | As-cast | 2.40 | 0.15 ± 0.03 | 0.78 ± 0.09 | 1.02 ± 0.06 | 11.57 ± 0.46 | 11.63 ± 0.40 | 34.94 ± 0.48 | 32.49 ± 0.79 | 7.42 ± 0.38 |
|  | HIP + S * | 0 | - | - | - | - | - | - | - | - |
| P | As-cast | 3.50 | 0.13 ± 0.05 | 1.29 ± 0.08 | 0.95 ± 0.05 | 11.15 ± 0.41 | 11.95 ± 0.36 | 34.63 ± 0.43 | 32.11 ± 0.70 | 7.80 ± 0.34 |
|  | HIP + S | 0.35 | 0.08 ± 0.06 | 2.02 ± 0.07 | 0.59 ± 0.09 | 11.43 ± 0.07 | 12.82 ± 0.24 | 29.76 ± 0.33 | 30.90 ± 0.76 | 12.39 ± 0.53 |
| N | As-cast | 2.60 | 0.17 ± 0.02 | 0.91 ± 0.11 | 0.81 ± 0.08 | 11.19 ± 0.21 | 11.10 ± 0.21 | 34.54 ± 0.26 | 33.94 ± 0.35 | 7.40 ± 0.16 |
|  | HIP + S | 0.19 | 0.09 ± 0.06 | 1.71 ± 0.07 | 0.52 ± 0.11 | 11.48 ± 0.24 | 11.89 ± 0.24 | 30.15 ± 0.037 | 30.85 ± 0.38 | 13.99 ± 0.69 |
| NP | Pre-HIP | 2.10 | 0.17 ± 0.06 | 1.19 ± 0.12 | 0.67 ± 0.14 | 10.90 ± 0.70 | 11.29 ± 0.39 | 32.59 ± 0.55 | 34.10 ± 0.97 | 9.09 ± 0.49 |
|  | Pre-HIP + HIP + S | 0.14 | 0.16 ± 0.05 | 1.76 ± 0.11 | 0.53 ± 0.05 | 10.98 ± 0. 30 | 11.83 ± 0.20 | 30.97 ± 0.71 | 31.28 ± 0.88 | 12.62 ± 0.93 |

**Table 3.** Segregation ratio (SR) of alloying elements in the as-cast state.

| Mould | Ti | Nb | Mo | Fe | Cr |
|---|---|---|---|---|---|
| O | 2.28 | 3.70 | 1.49 | 0.76 | 0.86 |
| E | 2.37 | 3.75 | 1.25 | 0.79 | 0.85 |
| P | 2.57 | 3.95 | 1.40 | 0.78 | 0.88 |
| N | 2.04 | 2.73 | 1.37 | 0.78 | 0.82 |

**Table 4.** Grain size (mean with and length with standard deviation) in the as-cast state.

| Mould | Mean Width (mm) | Mean Length (mm) | Aspect Ratio | Morphology |
|---|---|---|---|---|
| O | 1.1 ± 0.35 | 3.4 ± 0.92 | 3.1 | Columnar |
| E | 1.0 ± 0.46 | 3.2 ± 0.97 | 3.2 | Columnar |
| P | 1.1 ± 0.33 | 3.3 ± 0.78 | 3.0 | Columnar |
| N/NP | 2.1 ± 0.34 | 3.4 ± 0.71 | 1.5 | Coarse, slightly columnar |

*3.3. LBW Varestraint Weldability Test Results*

Figure 6 shows the hot cracking behaviour observed in the five casting heats that was determined by LBW Varestraint test. Moulds O (low Si), P (high Si), and E (standard Si), which were cast without ceramic blanket and therefore solidified at quicker cooling rates, were tested at augmented strain levels from 1 to 8%. Slowly cooled heats (moulds N and NP) were only tested up to 4% augmented strain. Results show that in all casting heats, FZ cracking was much more prominent than HAZ cracking. In fact, TCL measured in FZ was more than 5–6 times longer than in HAZ. At 8% augmented strains, TCL determined in the HAZ ranged between 2.5 and 5 mm, whereas for the same testing conditions, TCL in FZ was between 20 and 30 mm. Therefore, it was observed that LBW Varestraint test mainly enhanced FZ cracking.

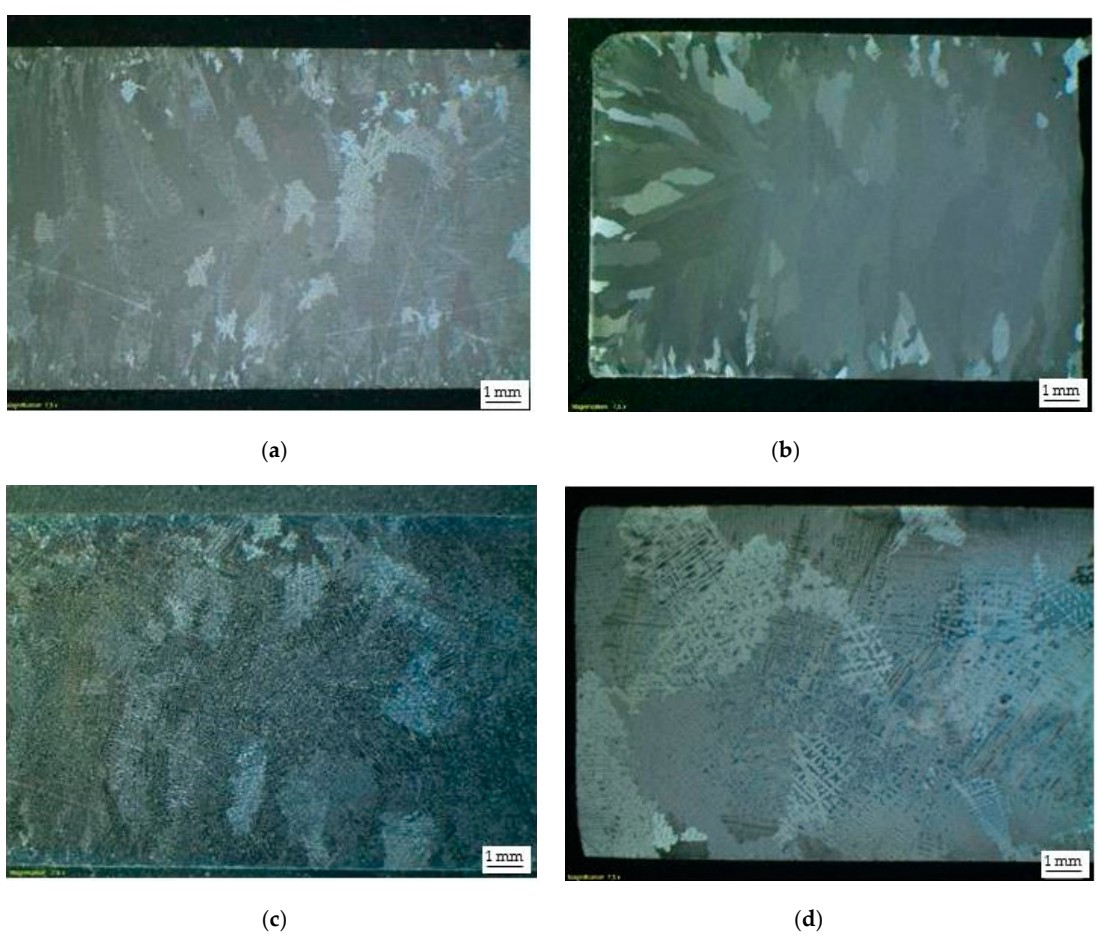

**Figure 5.** Grain size of different moulds in the as-cast state: (**a**) O, (**b**) E, (**c**) P, and (**d**) NP.

**Table 5.** Segregation ratio (SR) of alloying elements after heat treatment.

| Mould | Ti | Nb | Mo | Fe | Cr |
|-------|------|------|------|------|------|
| O | 1.29 | 1.37 | 1.28 | 0.89 | 0.93 |
| E | 1.27 | 1.16 | 1.32 | 0.94 | 0.95 |
| P | 1.20 | 1.34 | 1.28 | 0.93 | 0.94 |
| N | 1.13 | 1.52 | 1.52 | 0.98 | 0.99 |
| NP | 1.33 | 1.35 | 1.23 | 0.92 | 0.94 |

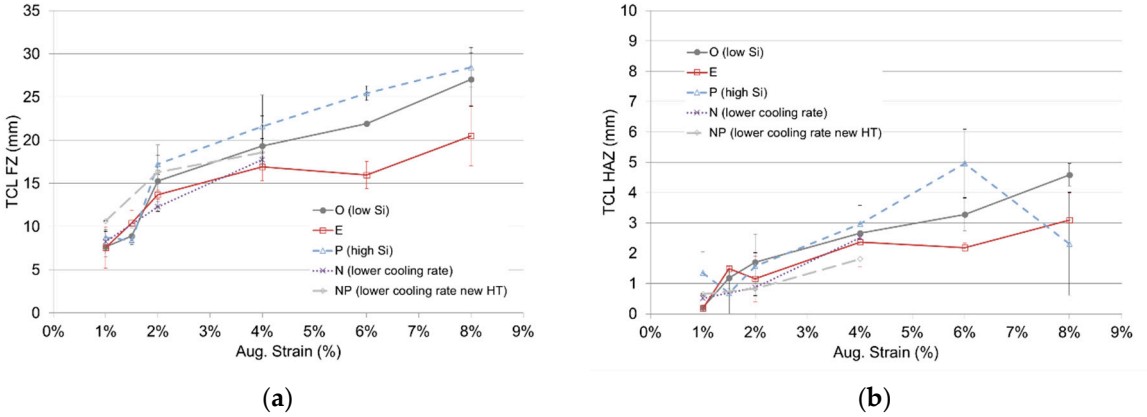

**Figure 6.** (**a**) FZ and (**b**) HAZ cracking response of five alloy 718 casting heats in LBW Varestraint test.

Mould P (high Si content) showed a comparatively higher cracking susceptibility than the rest of the heat levels. Nevertheless, mould O (low Si) was ranked second in terms of trend towards FZ and HAZ cracking. Mould E with intermediate Si content and NP showed slightly lower TCL values, whereas behaviour of mould N was in between.

### 3.4. Hot Ductility Weldability Test Results

Figure 7 describes the area reduction percentage determined in on-heating and on-cooling hot ductility tests for the five casting heats. Continuous lines correspond to the grade 3 polynomial fitting of experimentally determined on-heating test results, whereas dashed lines depict on-cooling behaviour. Looking at on-heating curves, the ductility at temperatures between 950 and 1000 °C was higher than 60% for the five moulds, showing comparatively higher values in the case of mould E and O (low Si content). Between 1000 and 1050 °C, ductility started to drop, and at 1150 °C it was already below 2%, which meant that the capability to deform without breaking had been completely lost. Again, at intermediate 1100 °C, moulds E and O (low Si content) showed comparatively better performance in terms of ductility, which was the reason why fitting curves were slightly displaced towards higher temperatures. This means that the onset of ductility drop in these two moulds was delayed to some extent.

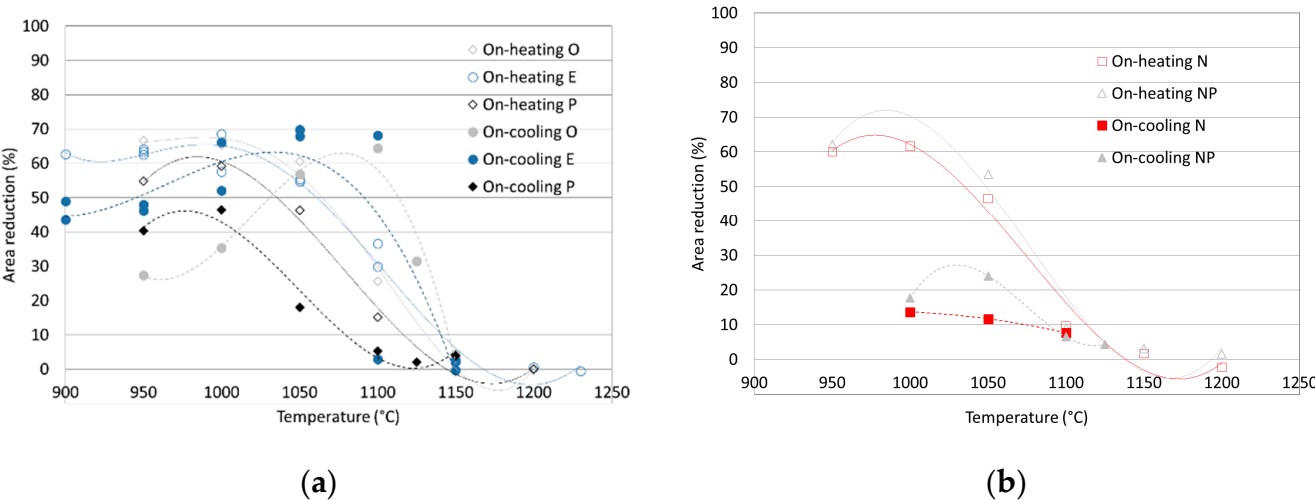

**Figure 7.** On-heating and on-cooling curves in terms of area reduction percentage and testing temperature for moulds E, O, and P (**a**), and N and NP (**b**).

Differences between casting heats were much more evident in on-cooling curves. Mould O (low Si content) presented remarkable ductility recovery behaviour, quickly reaching an area reduction value of 64% after testing at 1100 °C. Note that the thermal sequence of on-cooling hot ductility tests involved fast heating to peak temperature of 1195 °C and cooling down to the corresponding test temperature. Peak temperature was selected after defining NST in mould E samples that reached 1263.5 °C ± 5.8 °C. It was decided to limit peak temperature in on-cooling test to 1195 °C in order to ensure repeatable and stable ductility recovery behaviour.

After reaching ductility values or original parent material, mould O showed a ductility drop at lower testing temperatures down to 27% at 950 °C. This drop was not so remarkable either in mould E (from 69% to 46%) or high Si P (from 46% to 40%). Thus, it is quite clear that the ductility recovery rate is strongly related to the Si content of the alloys.

Moulds N and NP with slower cooling rates in the casting process depicted a completely different ductility recovery performance during on-cooling tests. In these two heats, restored ductility values did not surpass 14% and 24%, respectively, and the slope of the curves was drastically reduced. Calculated DRT and brittle temperature range (BTR) values are included in Table 6. BTR is the difference between peak temperature employed in

on-cooling tests and determined DRT, at which 5% of area reduction is recovered. BTR is a parameter widely used to conclude on the hot cracking susceptibility of superalloys [2,9,26].

**Table 6.** Weldability parameters obtained from hot ductility tests.

| Mould | Peak Temperature (°C) | DRT (°C) | BTR (°C) | Max. Ductility after Cooling | Ductility Recovery Rate |
|---|---|---|---|---|---|
| O | 1195 | 1150 | 45 | 64% at 1100 °C | Very fast |
| E | 1195 | 1145 | 50 | 69% at 1050 °C | Fast |
| P | 1195 | 1090 | 105 | 46% at 1000 °C | Intermediate |
| N | 1195 | 1110 | 85 | 14% at 1000 °C | Very low |
| NP | 1195 | 1110 | 85 | 24% at 1050 °C | Very low |

*3.5. Bead-on-Plate Weldability Test Results*

Figure 8 shows an image of the bead-on-plate tests performed on casting plates from five heats. Both straight and circular weld paths were applied. Circular welds were 15 mm in diameter and the goal was to modify the self-constraint condition. As described above, bead-on-plate tests were carried out both in 9 mm and less than 3 mm thickness plates, after surface grinding and EDM cutting, respectively, employing similar welding parameters as in LBW Varestraint tests. In the case of thinner samples, laser power was decreased to 2050 W to avoid excessive root overhang.

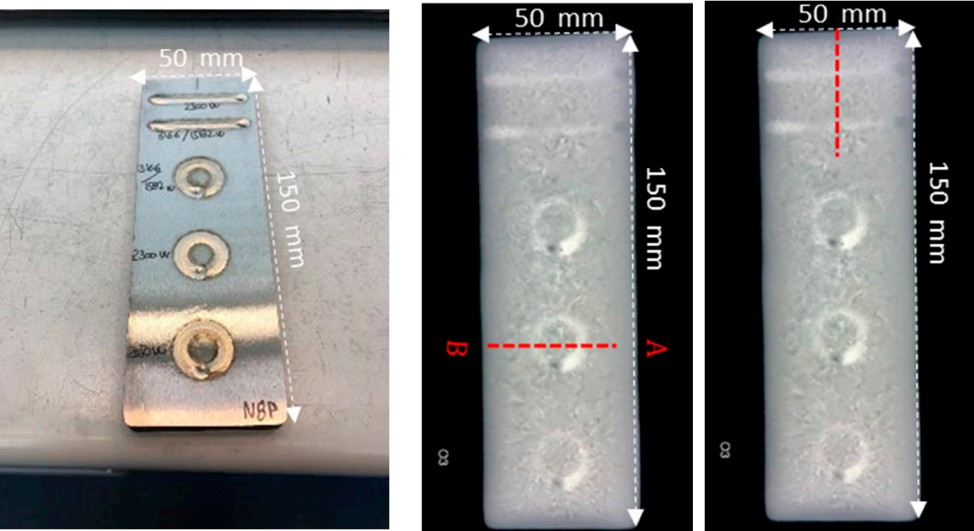

**Figure 8.** Bead-on-plate trials and X-ray digital images showing cuts for cross-section metallographic examination.

Every examined weld was free of cracks in less than 3 mm thickness samples, showing a "bowl-like" welding shape (Figure 9), that is, a shape without characteristic nail head of keyhole mode LBW. However, several cracks were detected in the HAZ in 9 mm thickness samples (Figure 10) welded with similar process parameters (2300 W), following both straight and circular welding paths. Note that in this case, welding resembled "nail or mushroom shape" usually observed in keyhole mode LBW.

Total number of cracks and TCL determined in each cross section were included in Figure 11. For circular welds, average values determined from cross sections A and B were represented. It is clearly shown that straight welds gave rise to longer and higher number of cracks with minor influence of alloy composition.

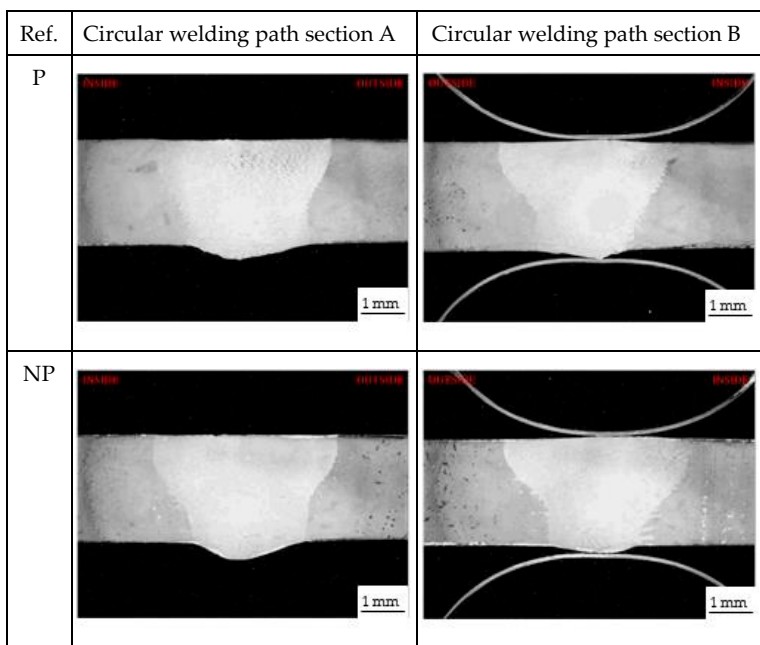

**Figure 9.** Cross-section of bead-on-plate tests of moulds P and NP with less than 3 mm thickness. Laser power 2050 W.

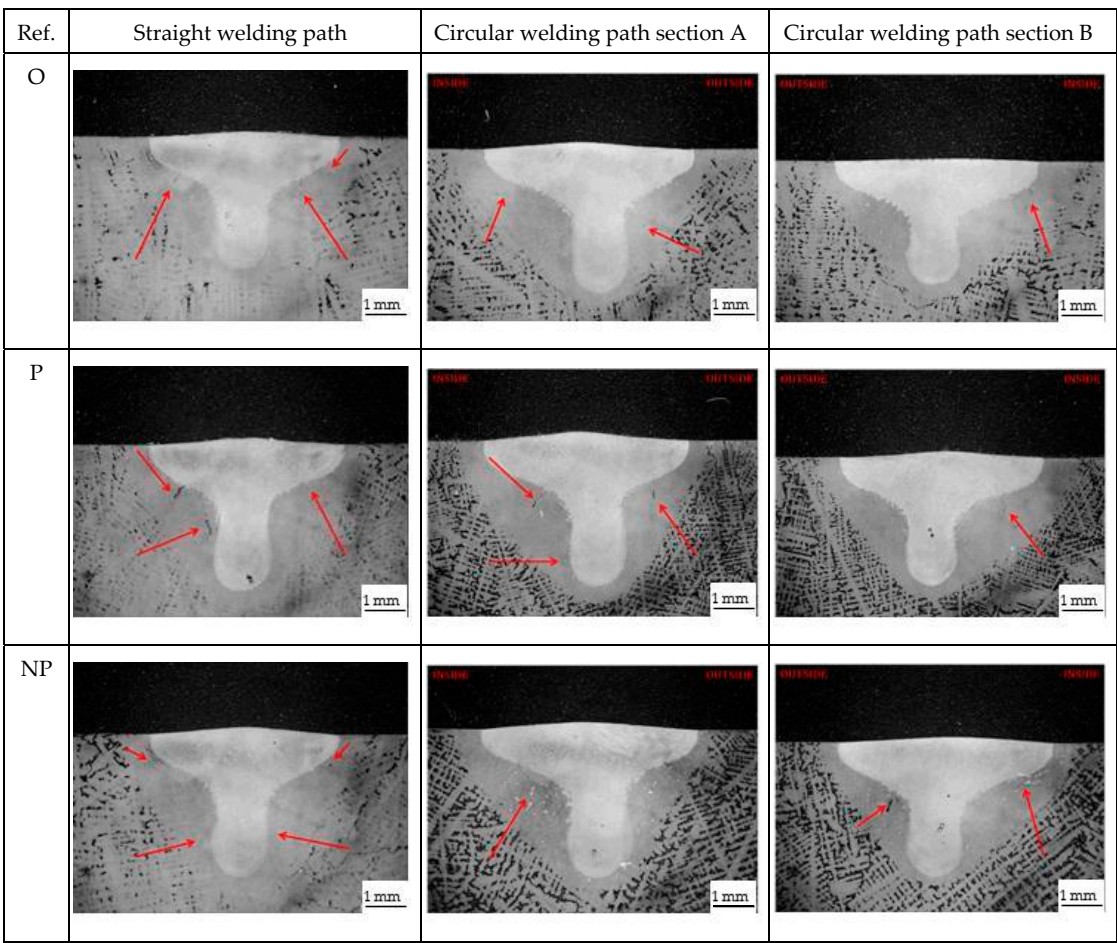

**Figure 10.** Cross-section of bead-on-plate tests of moulds O, P, and NP with 9 mm thickness. Laser power 2330 W.

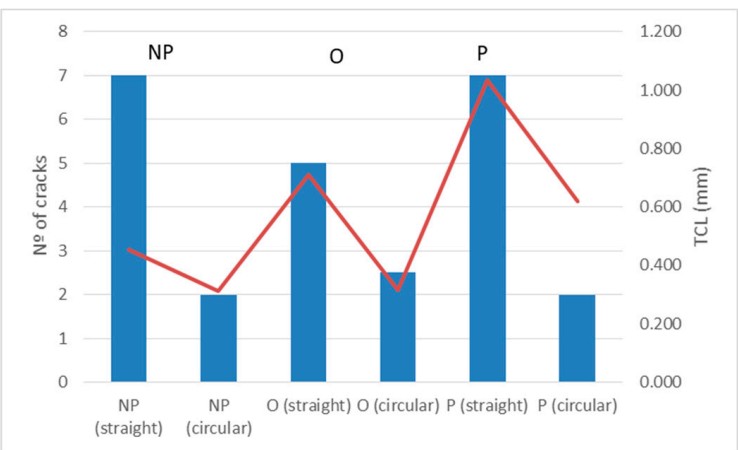

**Figure 11.** Number of cracks (blue bars) and TCL (red line) in bead-on-plate cross-sections of moulds O, P, and NP with 9 mm thickness. Laser power 2330 W.

*3.6. Microstructural Characterisation of Bead-on-Plate Welding Samples*

The composition of Laves phases in FZ and HAZ/BM (base metal) was analysed by EDX in both 9 and less than 3 mm thickness plates; corresponding values are included in Table 7. It can be observed that Laves phases in FZ had much lower Si, Nb, and Mo contents with respect to the HAZ and BM, and, on the contrary, they were enriched in Ti, Cr, Fe, and Ni elements. The Si content of Laves phase in FZ increased with the Si weight percentage of the alloy, with 0.22% being the lowest in mould O heat. In FZ the area percentage of Laves phases increased with increasing Si content of the alloy. Surprisingly, the area percentage of Laves phases of 3 mm plates of both moulds (P and NP) were lower than in the 9 mm thickness plates. The 9 mm thickness plates evacuated heat more quickly by thermal conduction, and therefore comparatively higher solidification rates and less Laves phases were expected.

**Table 7.** Area percentage and mean chemical composition of Laves phase (in wt %) and standard deviation in FZ and HAZ/BM of bead-on-plate welding samples.

| Mould | Location | Area % Laves | Al | Si | Ti | Cr | Fe | Ni | Nb | Mo |
|-------|----------|-------------|----|----|----|----|----|----|----|----|
| | | | | | 9 mm | | | | | |
| O | FZ | 1.78 | 0.41 | 0.22 | 1.51 | 14.02 | 14.80 | 43.52 | 20.53 | 4.99 |
| | HAZ/BM | - | - | - | - | - | - | - | - | - |
| P | FZ | 4.13 | 0.40 | 0.65 | 1.68 | 13.44 | 14.19 | 44.12 | 20.51 | 5.02 |
| | HAZ/BM | 0.20 | - | 1.98 | 0.53 | 11.16 | 12.55 | 30.70 | 30.33 | 12.74 |
| NP | FZ | 2.41 | 0.56 | 0.47 | 1.57 | 13.53 | 13.42 | 43.65 | 22.11 | 4.68 |
| | HAZ/BM | 0.32 | - | 1.55 | 0.57 | 11.64 | 12.24 | 31.20 | 29.14 | 13.66 |
| | | | | | Less than 3.0 mm plate | | | | | |
| P | FZ | 2.63 | 0.51 | 0.33 | 1.36 | 14.34 | 15.17 | 43.29 | 20.17 | 4.82 |
| | HAZ/BM | 0.20 | 0.14 | 2.01 | 0.5 | 10.96 | 12.43 | 30. 92 | 30.42 | 12.58 |
| NP | FZ | 1.50 | 0.48 | 0.49 | 1.46 | 13.61 | 13.88 | 43.07 | 22.07 | 4.94 |
| | HAZ/BM | 0.30 | 0.40 | 1.43 | 0.81 | 11.94 | 12.03 | 35.76 | 26.73 | 11.17 |

Laves phase was not detected in HAZ and BM of the low Si O alloy. In the high Si alloy (mould P) and the standard Si alloy with low solidification rate and pre-HIP (NP), small quantities of Laves phases in HAZ/BM were still observed, which were higher in the latter. This indicates that the time and temperature of additional heat treatment before HIP were not enough to completely dissolve these phases.

As described above, cracks in HAZ were only observed in the 9 mm thickness plates along the grain boundaries. Higher magnification SEM images are displayed in Figure 12.

This micrograph corresponded to mould P (high Si) and it was representative of the rest of the observed cracks in 9 mm thickness samples. These cracks were partially filled with a continuous Laves phase film, which was observed either at the tip of the crack or at the edges of the open crack. The chemical composition of this Laves film matched exactly with the Laves phase composition in FZ (Table 8).

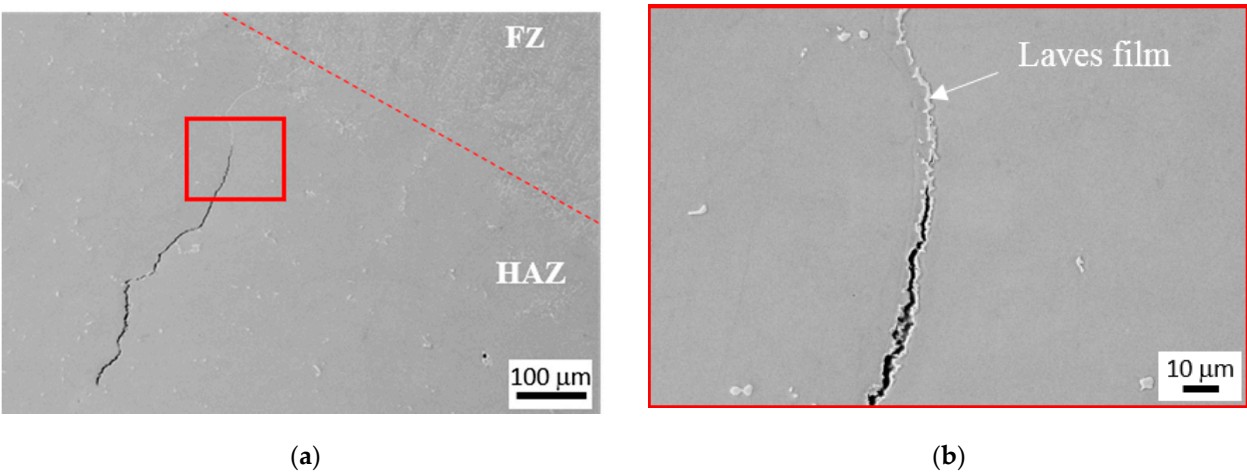

(**a**)                                                                                     (**b**)

**Figure 12.** (**a**) Grain boundary cracking in HAZ and (**b**) higher magnification of squared zone. Mould P.

**Table 8.** Composition of Laves phases (in wt %) at different locations in 9 mm thickness sample of Mould P.

| Laves Phase | Al | Si | Ti | Cr | Fe | Ni | Nb | Mo |
|---|---|---|---|---|---|---|---|---|
| Crack-HAZ | 0.83 | 0.73 | 1.45 | 12.93 | 12.94 | 42.36 | 23.68 | 5.08 |
| FZ | 0.40 | 0.65 | 1.68 | 13.44 | 14.19 | 44.12 | 20.51 | 5.02 |
| HAZ | - | 1.98 | 0.53 | 11.16 | 12.55 | 30.70 | 30.33 | 12.74 |

Besides cracks, there was also evidence of Laves phase liquation observed in the HAZ of moulds P and NP for both less than 3 mm and 9 mm thickness plates. These phases were located in the interdendritic region, also showing the presence of carbides and δ phase (Figure 13).

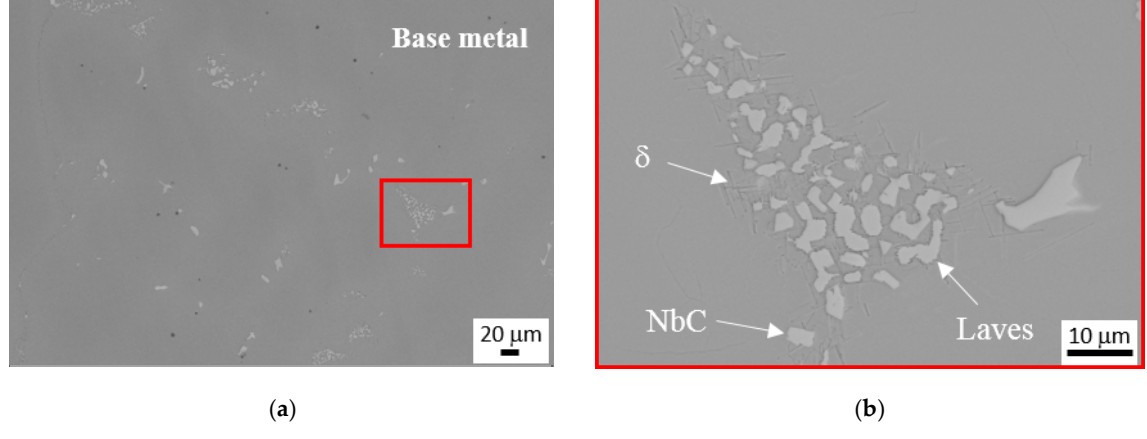

(**a**)                                                                                     (**b**)

**Figure 13.** *Cont.*

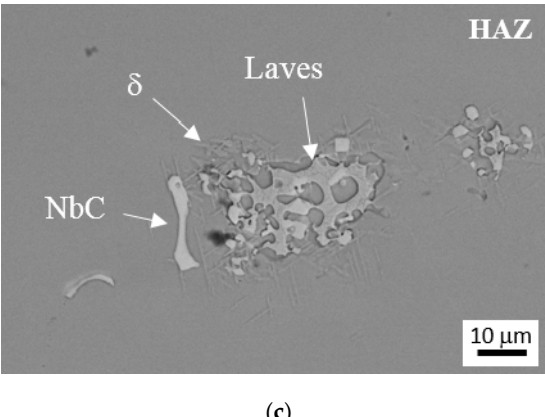

(**c**)

**Figure 13.** (**a**) Phases in the interdendritic region. (**b**) Magnification of the squared zone showing the presence of δ phase, carbides, and Laves phases in base metal (BM) in the interdendritic segregation area. (**c**) Segregation area with evidence of Laves and δ phase liquation in HAZ (NP- less than 3 mm plate).

## 4. Discussion

### 4.1. Influence of Chemical Composition, Investment Casting Conditions, and Heat Treatment on Microstructure and Weldability

Moulds E and O with higher casting solidification rates (1.65 °C/s) and standard and low Si contents, respectively, did not present any Laves phases in solution annealing state after complete HIP + solubilisation annealing heat treatment. Area percentage of Laves phase observed in moulds N, NP, and P in the solution-annealed condition ranged from 0.14 to 0.35%, without significant differences between moulds N and NP and being the residual content of Laves phase higher in mould P, that is, the heat with higher Si content and faster solidification rate. Note that this mould also had the highest amount of Laves phase in the as-cast condition (Table 2).

In moulds P, N, and NP containing residual Laves phase in the solution annealing state, the onset of ductility drop in on-heating hot ductility tests was triggered at lower temperatures, and corresponding fitting curves were shifted to the left side of the chart. This was observed in curves displayed in Figures 7a and 14, with the latter superposing the on-heating curves of the five studied heats. Note that dashed fitting lines corresponded to those three moulds. Therefore, this early ductility drop was associated with incipient melting of Laves phase. It is worth mentioning that melting of Laves phase is a fast event which does not require any significant reaction time to form the liquid as in constitutional liquation of NbC [2,6], and, consequently, it readily melts upon heating at very fast rates, as in current hot ductility tests (111 °C/s heating rate).

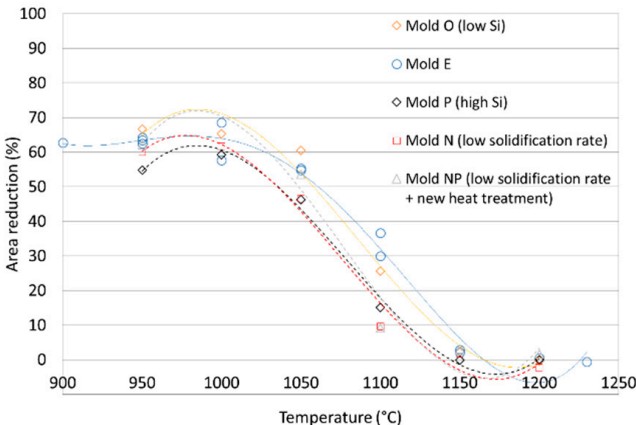

**Figure 14.** Comparison of on-heating curves in terms of area reduction percentage and testing temperature for moulds E, O, P, N, and NP.

Effective dissolution of Laves phase in alloy 718 castings prior to welding and high temperature use is pursued in industrial manufacturing processes because this phase impairs both weldability and mechanical properties [4,5]. This has been particularly shown in castings with volume fractions of secondary particles (including Laves and NbC) higher than the residual values reported in this work.

The chemical composition of remaining Laves phase was remarkably modified in moulds P, N, and NP after HIP and solubilisation treatment. The considerable enrichment in Mo could be explained by differences in diffusivities of solute elements in the austenite matrix. The dissolution kinetics of Laves phase in IN718 has been studied by [28]. In this work, the authors concluded that the back-diffusion of molybdenum in austenite is the controlling micro-mechanism for dissolution of the undesirable Laves phase. On the basis of the Johnson–Mehl–Avrami–Kolmogorov (JMAK) analysis at different temperatures, the authors determined the activation energy of 274.5 kJ/mol for the dissolution of Laves phase, which was close to the activation energy for diffusion of Mo in Ni (288 kJ/mol). This was also supported by the diffusion calculations shown in Table 9, which shows lower diffusivity values for Mo in Ni than for Nb and Ti.

**Table 9.** Parameters for diffusivity calculation of Nb, Ti, and Mo in Ni and calculated diffusivities.

| Element | Q (kJ/mol) | Do ($m^2$/s) | $D_{1050\,°C}$ ($m^2$/s) | $D_{1100\,°C}$ ($m^2$/s) | $D_{1150\,°C}$ ($m^2$/s) | Reference |
|---------|-----------|--------------|------------------------|------------------------|------------------------|-----------|
| Nb | 202 | $1.0 \times 10^{-6}$ | $1.1 \times 10^{-14}$ | $2.1 \times 10^{-14}$ | $4.0 \times 10^{-14}$ | [29] |
| Ti | 257 | $86 \times 10^{-6}$ | $0.6 \times 10^{-14}$ | $1.4 \times 10^{-14}$ | $3.2 \times 10^{-14}$ | [30] |
| Mo | 288 | $300 \times 10^{-6}$ | $0.1 \times 10^{-14}$ | $0.8 \times 10^{-14}$ | $0.8 \times 10^{-14}$ | [31] |

It is also interesting to note that the chemical composition of remaining Laves phase in these three heats was comparable with only minor differences in Si content.

HIP and solubilisation treatment did not modify original grain size and morphology resulting from investment casting. Consequently, different cooling conditions during casting yielded different grain sizes and aspect ratios which remained in the base material employed for the weldability tests. Thus, moulds O, E, and P cooled without ceramic blanket had highly columnar grains elongated along plate thickness. On the contrary, HIP and subsequent solution annealing heat treatment was effective in reducing segregation in interdendritic regions, leading to comparable segregation ratios in the five heats (Table 5).

*4.2. Correlation between Weldability Assessment Trials*

In LBW Varestraint test, which is an externally loaded weldability test, cracks were mainly induced on the surface and in FZ since the strain was applied while the melt pool was solidifying and it was forced to pull away when the material did not have minimum strength and ductility to accommodate residual deformations. FZ cracking was highly enhanced in LBW Varestraint test, featuring an elongated V-shape solidification line due to the LBW parameters and energy density which was required to achieve full penetration and minimum weld width requested by industrial quality standards [16]. In this study, the section of Varestraint testing samples had to be reduced to 3.2 mm to deform them by bending due to test bench capability and to adjust them to representative welding applications.

Minimum differences in HAZ cracking were determined between heats with somehow better performance of NP and E. However, the relatively high scattering of TCL va-lues made the comparison between alloys difficult.

Clear differences between alloys were observed in on-cooling hot ductility Gleeble tests. In these tests, only incipient melting of low melting point phases took place. This melting was enough to wet grain boundaries, leading to a brittle fracture without any



area reduction at the test temperatures close to NDT. By reducing on-cooling testing temperature, we were able to study ductility recovery behaviour. On-cooling hot ductility results showed that moulds O and E (without Laves phase in BM at the beginning of the welding test) had shorter BTR and fast ductility recovery rates reaching the previous values before later ductility drop (Table 6). In both moulds, the ductility was effectively recovered at temperatures above 1050 °C (Figure 7). Moulds N and NP presented higher BTR (85°) and very low recovery rate and ductility recovery capability. Coarser γ grain sizes in these samples could be the reason for this behaviour, since large grain sizes promote continuity of liquid resulting from incipient melting at grain boundaries and reduce extension of interfacial area between solid-state γ grains [2].

Mould P with higher Si content in its chemical composition had the largest BTR (105 °C) and intermediate recovery rate and capability. Longer BTR can be related to both the higher amount of residual Laves phase in the microstructure and its greater Si content. Both factors will contribute to increasing the volume fraction of intergranular liquid formed at the peak temperature reached in on-cooling test. Consequently, the temperature must be decreased to a lower point to allow full resolidification of the Si-enriched liquid. It must be mentioned that the effect of Si content on 718 alloys has been previously investigated, concluding that HAZ cracking trend is favoured if high Si contents are combined with high Mn or C contents [2,4].

Hot ductility tests give an insight about the response of material to liquation and subsequent resolidification. However, only a very limited amount of material is melted as opposed to real welding applications in which a relatively high amount of material is melted in the FZ and significant microstructural changes occur in this zone.

Indeed, this was observed in the FZ microstructural characterisation of bead-on-plate samples. Laves phases were regenerated in FZ, whose chemical composition was completely different from the original Laves phase in HAZ and BM. FZ Laves phase in 9 mm plates were particularly depleted in Nb, Si, and Mo (Table 7). Surprisingly, the number of Laves phases in the less than 3 mm plates were much lower than in the 9 mm plates with faster welding cooling rates.

Whereas 3 mm plates were free of cracks, remarkable microfissures were identified in HAZ along grain boundaries in 9 mm bead-on-plate samples for the five investigated heats. As described previously, these cracks were decorated by continuous film, whose composition matched with Laves phase of FZ (Nb (20.5–22.1 wt %), Mo (4.7–5.0 wt %), and Ti (1.5–1.7 wt %)). Composition of this Laves phase film is quite independent of chemical composition, with only minor deviations in Si content. Laves phase with very similar Nb concentrations in LBW welds were reported by Odabaçi et al. [25].

As can be observed in Figures 12 and 15, corresponding to cross-sections of 9 mm thickness bead-on-plate samples of moulds P and O, respectively, the Laves film was also extended through FZ. This is strong evidence of backfilling mechanism. The current results demonstrate that in 9 mm bead-on-plate samples, the terminal liquid which remains in the melt pool at the end of the solidification process was diffused along grain boundaries, giving rise to a continuous Laves phase film that caused HAZ microfissuration.

Looking at the micrographs, we concluded that this diffusion was completed along base material γ grain boundaries which are perpendicular to fusion line and delimit three-point intersections. Similar crack morphologies have been reported by Bai et al. [19] in LBW. The area below the nail head is a particular critical point where these microfissures pop up. It is worth mentioning that grain morphology was quite columnar, with grain boundaries elongated along the thickness, particularly in samples cast without ceramic blanket. Therefore, at the nail head area where the fusion line was quite horizontal, there was a high probability of having grain boundaries in the parent material intersecting fusion line, leading to three-point intersection, which is a critical point with high stresses during melt pool solidification [2]. Moreover, the elongated and straight morphology of vertical grain boundaries would be favourable for the described diffusion-based backfilling mechanism. Evidence of comparable backfilling mechanism which promotes HAZ cracking

has been recently reported in Haynes® 282® casting alloy [32]. In this case, cracking was exacerbated by diffusion of B. Other authors cited by [2] also concluded that segregation of B has a detrimental effect of HAZ liquation and FZ solidification cracking. However, this is not an influencing parameter in this work since ingots with the same lean B content of 0.002 wt % were used as raw material for the five heats.

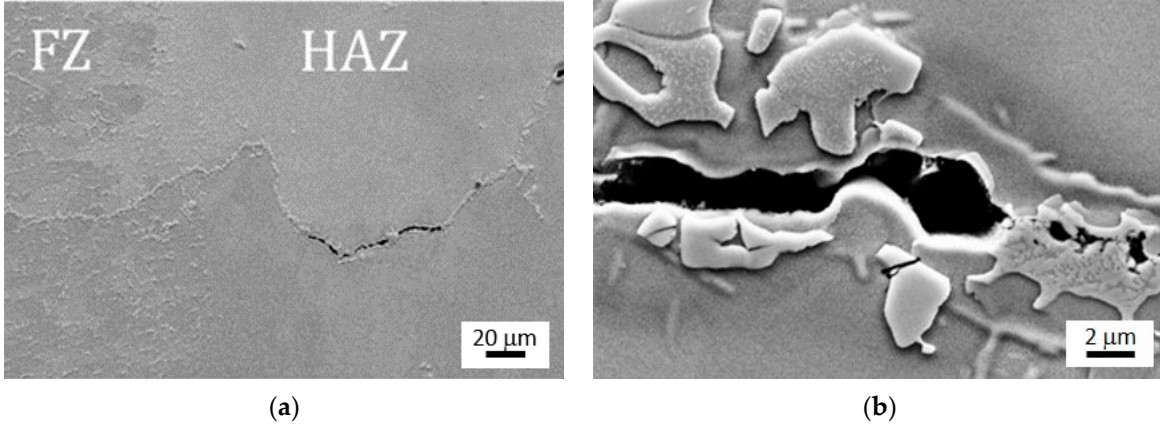

**Figure 15.** (**a**) Film along crack in HAZ from 9 mm thickness sample mould O cross-section. (**b**) Enlargement of crack tip.

Straight bead-on-plate welds were more prone to HAZ cracking phenomena since both total number of cracks observed in cross-sections and TCL of those cracks were comparatively higher than in circular welds (Figure 11). Bead-on-plate samples from NP mould showed a consistently lower TCL than equivalent samples from moulds O and P. Therefore, it can be concluded that the lower aspect ratio of these grains due to slower solidification rates during initial casting reduced the probability of having deleterious three-point intersections along the length of fusion line.

No cracks were observed in less than 3 mm thickness samples welded with comparable LBW parameters. In this case, two remarkable differences were observed when looking at the cross-section of these samples. On one hand, melt pool or FZ had a "bowl shape" without nail head, typical in keyhole mode LBW, which reduces the risk of perpendicular intersection with columnar grains of parent material in comparison with "nail or mushroom" shape observed in 9 mm thickness plates. On the other hand, percentage of Laves phases in FZ was much lower than in 9 mm plates, as observed from Table 7. Both factors were critical to avoid formation of HAZ microfissures observed in thicker plates.

The percentage of Nb content of Laves phase observed in FZ of bead-on-plate samples was close to the eutectic point of the pseudo-binary diagram of alloy 718 (Figure 16a). This means that the terminal liquid would solidify as L → γ + Laves eutectic. It must be noted that most Laves phases observed at high magnification had a eutectic microstructure composed of γ and Laves phase (Figure 16b). This solidification path would be associated with larger volumes of terminal liquid that solidified at lower temperatures (down to 1180 °C) [4], since stepped liquid to solid transformation is hampered. If the amount of terminal liquid and its coexistence temperature range are increased, then one should expect a higher grain boundary wetting and infiltration risk in the three-point intersections, particularly if those grain boundaries are long and straight, as in the case of columnar microstructures obtained in the investment casting process.

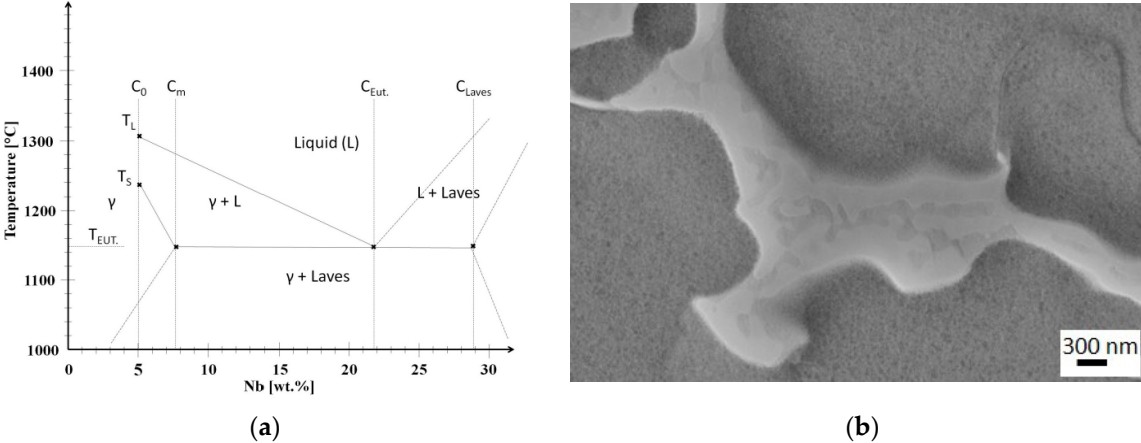

(**a**)                                              (**b**)

**Figure 16.** (**a**) Pseudo-binary phase diagram for alloy 718 [33][1] and (**b**) eutectic Laves phase in FZ (mould NP). [1] Reproduced from [33], with permission from J. Andersson, 2021.

In order to analyse the solidification path and steps, we carried out thermodynamic and diffusion-based simulations with the three heats with different Si contents. Figure 17a shows the calculated solidification path of mould E determined using Scheil simulation, which considers back diffusion of elements in the primary phase at high cooling rate of 100 °C/s, representative of welding processes. The simulation predicted the formation of NbC carbides and Laves phase at the final stage of solidification, significantly reducing the solidus temperature compared to solidification under equilibrium condition. In Figure 17b, we can see that at the end of solidification, the liquid enriched in Nb, Mo, Ti, and Si, which allowed the formation of Carbides and Laves phases, lowering the solidus temperature. Table 10 depicts the solidus and liquidus temperatures of the heats with different Si contents. It can be observed that the solidus temperature decreased with increasing Si content and the solidification range increased. As observed experimentally, the percentage of Laves phase and its Si content increased in the as-cast material through increasing the Si content of the alloy. However, in LBW, the cooling rate can be extremely fast [25] and this leads to the limitation of Nb segregation. Therefore, the resulting terminal interdendritic liquid will have a composition close to the eutectic point, and therefore it will have long persistence and relatively high volume.

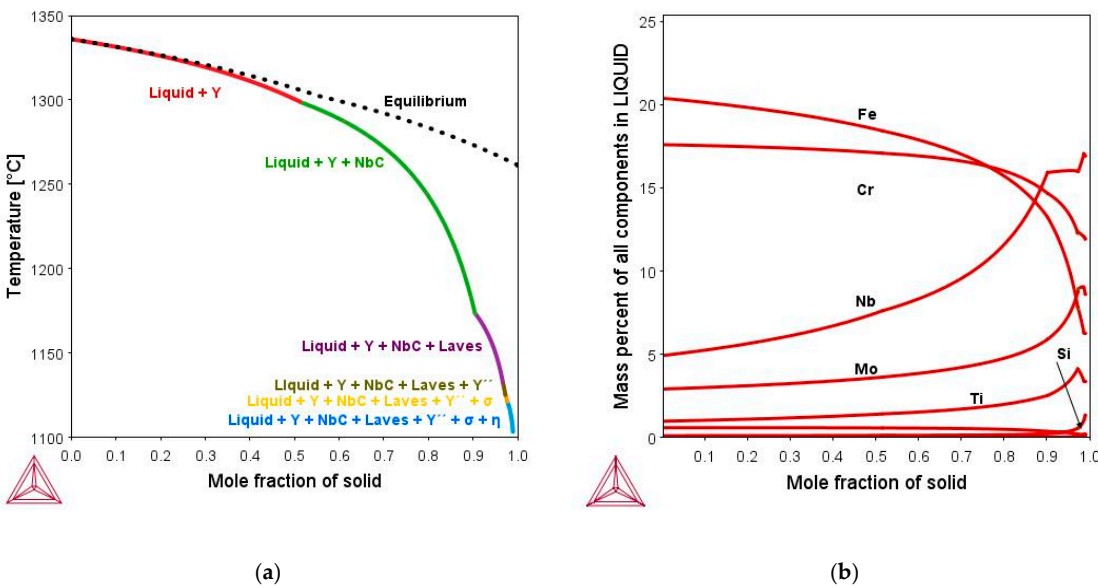

(**a**)                                              (**b**)

**Figure 17.** Thermo-Calc simulation of alloy E (standard Si content): (**a**) solidification path and (**b**) microsegregation prediction using the Scheil model.

**Table 10.** Solidus and liquidus temperatures determined by equilibrium and Scheil simulation.

| Ref. | Si in Alloy (wt %) | Si in Laves (wt %) | Laves wt % | Equilibrium Simulation | | | Scheil Simulation | | |
|------|------|------|------|------|------|------|------|------|------|
| | | | | Tsol (°C) | Tliq (°C) | ΔT (°C) | Tsol (°C) | Tliq (°C) | ΔT (°C) |
| O | 0.051 | 0.03 | 1.71 | 1213.6 | 1339.8 | 126.2 | 1110.1 | 1339.8 | 229.8 |
| E | 0.110 | 0.13 | 1.78 | 1212.0 | 1335.8 | 123.8 | 1103.5 | 1335.8 | 232.4 |
| P | 0.170 | 0.43 | 1.91 | 1209.2 | 1338.0 | 128.8 | 1084.0 | 1338.0 | 254.1 |

## 5. Conclusions

In this work, influence of Si content, solidification rate, and pre-weld heat treatment on as-cast microstructure, weldability, and hot cracking susceptibility of alloy 718 investment castings were investigated. The following conclusions can be drawn about the impact of these factors and the selection of the different weldability tests which have been employed:

- Microstructural analysis of as-cast samples showed differences between heats in terms of amount and chemical composition of Laves phase, grain size, and aspect ratio. Shape of γ grains mainly depended on cooling rate.

- After HIP and solution annealing heat treatment, residual contents (less than 0.35% in area) of Laves phase were only observed in the samples with higher Si content and slower solidification rate, i.e., moulds P, N, and NP. The application of additional pre-HIP cycle to slowly solidified casting at 1052 °C for 2 h was not enough to completely remove the Laves phase.

- Onset of hot ductility drop in on-heating hot ductility test was directly related to the presence of residual Laves phase, whereas the hot ductility recovery behaviour was connected to the Si content and parent material grain size. Coarser grain size was associated with very slow recovery rate and very limited ductility recovery capability due to longer liquid continuity after incipient melting. In parallel, higher Si content reduced DRT and enlarged BTR.

- LBW Varestraint tests gave rise to enhanced fusion zone (FZ) cracking with much reduced heat-affected zone (HAZ) cracking on the surface. Both TCL FZ and TCL HAZ were mostly independent of Si content and presence of residual Laves phase.

- In all LBW welds, Laves phase was formed again in the FZ. The chemical composition of regenerated Laves phase has a composition similar to the eutectic of the pseudo-binary equilibrium diagram of alloy 718, and this suggests a long persistence of terminal liquid during the welding solidification. FZ Laves phase had eutectic morpho-logy.

- The composition of the regenerated FZ Laves phase matched with the continuous Laves phase film observed along HAZ microfissures in LBW bead-on-plate samples with nail or mushroom shapes which are characteristic in keyhole mode LBW.

- The observed HAZ cracking can be explained by the following hot cracking mechanism: backfilling and infiltration of terminal liquid along parent material γ grain boundaries in three point intersections resulting from perpendicular crossing of columnar grain boundaries with fusion line. This cracking mechanism was enhanced by both nail or mushroom weld shapes and narrow and columnar grain sizes of castings.

- The described cracking mechanism did not depend on the Si content, the effective dissolution of Laves phase, or homogenization of segregation gradients with proper heat treatments before welding, since the formation of detrimental Laves phases happens in the final solidification of melt pool after welding, which takes place at very fast cooling rates and limits Nb segregation in comparison with slow cooling condition.

- Neither Varestraint nor hot ductility weldability tests can reproduce this particular cracking mechanism, which is activated inside the samples and requires remelting of significant amount of material to form the melt pool.

**Author Contributions:** P.Á.: Investigation in Varestraint, hot ductility, and bead-on-plate tests; methodology; formal analysis; and writing—original draft preparation. L.V.: Investigation in Varestraint, hot ductility, and bead-on-plate tests; methodology; data curation; and visualisation. A.C.: Investigation in Varestraint, hot ductility, and bead-on-plate tests; microstructural investigation; and visualisation. N.R.: Methodology, formal analysis, Varestraint investigation, and data curation. P.P.R.: Conceptualisation, methodology, investigation in investment casting, and writing—review. A.N.: Methodology, formal analysis, thermodynamic simulation, microstructural investigation, data curation, and writing—original draft preparation. A.M.: Methodology, formal analysis, microstructural investigation, and data curation. F.S.: Conceptualisation, methodology, supervision, writing—review and editing, project administration, and funding acquisition. All authors have read and agreed to the published version of the manuscript.

**Funding:** This research was performed under the framework of HiperTURB project, which was funded by Clean Sky 2 Joint Undertaking under the European Union's Horizon 2020 research and innovation program, grant agreement no. 755561.

**Institutional Review Board Statement:** Not applicable.

**Informed Consent Statement:** Not applicable.

**Data Availability Statement:** Data available on request due to restrictions. The data presented in this study are available on request from the corresponding author. The data are not publicly available due to some IPR and confidentiality issues.

**Acknowledgments:** Bengt Pettersson and Vikström Fredrik from GKN Aerospace company in Trollhättan (Sweden) are gratefully acknowledged for their technical support and fruitful discussions. Additionally, the authors would like to thank the support received from Kejll Hurtig and Joel Andersson (Department of Engineering Science at West University) in carrying out hot ductility tests.

**Conflicts of Interest:** The authors declare no conflict of interest.

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
