# Peer review of "Weldability Evaluation of Alloy 718 Investment Castings with Different Si Contents and Thermal Stories and Hot Cracking Mechanism in Their Laser Beam Welds"

_metals, doi:10.3390/met11030402_

Round 1

Reviewer 1 Report

The paper is not well written at this stage. It is not possible to include the journal without significant revision of the paper carefully. The followings are the comments to authors.

  1. Abstract is too long.
  2. Line 104, Ductility Recovery Temperature (DTR) should be DRT.
  3. Scale bar in the photographs of each figure is not possible to read.
  4. Magnification of micrographs is not recommended to include the manuscript.
  5. Table 3, "segregation ratio"; definition and the procedure should be explained. Table 5, the same.
  6. Table 6, "BTR", nothing is explained.
  7. Fig.10; "N8P, O3, P23", nothing explained. TCL must have unit. Blue bars and Red Lines, which data?
  8. Fig. 12 is appeared before Fig.11 and appeared again after Fig.11.
  9. English and other detailed explanations are recommended to be checked carefully because the reviewer found many mistakes and not English words.

Author Response

  1. Abstract is too long. A shorter version is provided.
  2. Line 104, Ductility Recovery Temperature (DTR) should be DRT. Two wrong entries changed.
  3. Scale bar in the photographs of each figure is not possible to read. Scale bars were changed.
  4. Magnification of micrographs is not recommended to include the manuscript. Magnifications from SEM images were eliminated.
  5. Table 3, "segregation ratio"; definition and the procedure should be explained. Table 5, the same. Definition and procedure to determine it were included.
  6. Table 6, "BTR", nothing is explained. Description and explanation of this parameter was introduced.
  7. Fig.10; "N8P, O3, P23", nothing explained. TCL must have unit. Blue bars and Red Lines, which data? References to individual samples taken from moulds deleted, unit of TCL and explanation of bars and line included.
  8. Fig. 12 is appeared before Fig.11 and appeared again after Fig.11. Caption of this figure was wrong.
  9. English and other detailed explanations are recommended to be checked carefully because the reviewer found many mistakes and not English words. Paper has been reviewed by native speaker. No English words have been deleted.

Reviewer 2 Report

  The manuscript analyzed the weldability and hot cracking susceptibility of 718 alloy. The author presented a number of experimental methods. However, the ‘Introduction’ lacks data on laser welding 718 alloy. What are the significant benefits of the experimental results? So I suggest that the manuscript needs some revisions before publication.

  • The Abstract was too long.
  • The ‘% of Laves phase and its Si content’ was wrong, the ‘%’ could be replaced by ‘percentage’.
  • Line 123-124. The statements about references [19-24] were too few. Please explain the significance of these references.
  • The welding parameters should be given in the manuscript.
  • Line 236-238. “the amount of Laves increases from 2.2 % to 3.5 %” The unit of numbers should be added. And please check the whole manuscript.
  • How do you identify the type of phases in Fig. 3? The morphology of Laves and Carbide was very similar in Fig. 3(b, c, d). And the type of phases in Fig. 3(c) was not labeled.
  • In Fig. 9. What were these red arrows pointing at? Cracks? It was hardly to observed in the original figure. The corresponding area in figure should be magnified.
  • Line 370-371. What was the captions of these two figures ‘(a) and (b)’, and the typography of these two figures was also wrong.
  • The unit of composition of phases in Table should be added, wt% or at%? Please check the whole manuscript.
  • The scaleplate in figures was too small to see the numbers clearly.
  • The conclusion part is too long and should be further summarized and condensed.

Author Response

  • The Abstract was too long.                                                                       A shorter version is provided.
  • The ‘% of Laves phase and its Si content’ was wrong, the ‘%’ could be replaced by ‘percentage’.                                                                      We are referring to area percentage of Laves and wt% of Si. “The area percentage of Laves phases increases from 2.2 %to 3.5 % and the Si content in the Laves phases increases from 0.28 to  1.29 wt%.
  • Line 123-124. The statements about references [19-24] were too few. Please explain the significance of these references.                     Explanation of significance was done.
  • The welding parameters should be given in the manuscript.         Parameters employed in LBW Varestraint tests are included in lines 174 to 176 of original manuscript. In lines 207 to 209 it is mentioned that the same welding parameters were used for Bead on plate tests.
  • Line 236-238. “the amount of Laves increases from 2.2 % to 3.5 %” The unit of numbers should be added. And please check the whole manuscript. The  amount is given as area percentage and the unit is %. A revision has been done through the whole manuscript.
  • How do you identify the type of phases in Fig. 3? The morphology of Laves and Carbide was very similar in Fig. 3(b, c, d). And the type of phases in Fig. 3(c) was not labeled.                                                                Explanation is included in the revised version.  Laves phases and carbides show similar colour and also morphologies when analysed by SEM, thus for identification it is necessary to perform an EDX analysis. Therefore, for the quantification of the Laves phase area percentage, first the area percentage of Laves phases + carbides was determined by SEM images. Second, the area fraction of carbides was evaluated by optical microscopy which reveals in the unetched state only the presence of carbides. Finally, the Laves phase area percentage was obtained by subtracting the area percentage of carbides obtained by OM from the area percentage obtained by SEM (Laves + carbides). 
  • In Fig. 9. What were these red arrows pointing at? Cracks? It was hardly to observed in the original figure. The corresponding area in figure should be magnified.                                                                                          Yes, they are cracks. Explanation has been included. The purpose of these images is to show weld morphology and clear indication that cracks mainly appear below the head of the nail. Higher magnification SEM images are given in other images.
  • Line 370-371. What was the captions of these two figures ‘(a) and (b)’, and the typography of these two figures was also wrong.                             Explanation was given and an additional image was introduced.
  • The unit of composition of phases in Table should be added, wt% or at%? Please check the whole manuscript.                                                Composition is wt%, amount of Laves phase is area percentage. The whole doc was checked.
  • The scaleplate in figures was too small to see the numbers clearly.            Scale bars were enlarged.
  • The conclusion part is too long and should be further summarized and condensed.                                                                                      Contents of conclusion section have been shortened.

Reviewer 3 Report

  1. Into introduction section there are too many conclusions, example line 37. This was strong evidence of backfilling mechanism which is described as wetting...." This information is very important, however should be transferred to discussion section.
  2. line 183-184 - R de radio of.... typo, correct
  3. line 202 if K-type thermocouple was welded to the surface it is measurement related to changing of temperature only on the surface, this type of measure is not reliable according to complex surface thermodynamic equilibrium according to radiation, forced convection and conduction phenomena. In the future it is recommended to use more than one thermocouples if only surface temperature changes was measured, or weld thermocouples to material in performed drilled hole
  4. line 329 it is more like U-groove weld not nail shape
  5. line 371 First Figure 12 is redundant because the same pictures was showed below line 375
  6. The authors of the article mention the interdendritic region, but there is no data that would clearly indicate this type of structure.
  7. The authors assume that their previous articles are known and omit some explanations. In my opinion, this should be avoided in the future.
  8. In welding processes, in particular laser welding, more talks about crystallization than solidification, because in this process we have a directional grain growth.
  9. Studying welding process based on obtained trial welds, some defects analysis are recommended, are there any welding defects beside cracks, for example porosity etc. is weld build is proper...

Article very interesting shoved extensile microstructure analysis based on Laves phase study. Recommended to publishing.

Author Response

  1. Into introduction section there are too many conclusions, example line 37. This was strong evidence of backfilling mechanism which is described as wetting...." This information is very important, however should be transferred to discussion section.                                                             A shorter abstract has been included in the new revision. I am in favour of keeping most important findings and conclusions in the abstract to call the attention of readers.
  2. line 183-184 - R de radio of.... typo, correct Done
  3. line 202 if K-type thermocouple was welded to the surface it is measurement related to changing of temperature only on the surface, this type of measure is not reliable according to complex surface thermodynamic equilibrium according to radiation, forced convection and conduction phenomena. In the future it is recommended to use more than one thermocouples if only surface temperature changes was measured, or weld thermocouples to material in performed drilled hole.                         This is the usual procedure to monitor and control temperature in hot ductility tests. Thermocouples are welded by resistance to the surface of the sample in the middle section, between clamps. There is no space to weld more than one thermocouple and it makes no sense, since, the temperature is homogeneous only in a short central area of the sample. The thermocouples might break or lose their measurement capability during the test. In that case, the test must be repeated. Samples cannot be drilled because the area in the calibrated length will change.
  4. line 329 it is more like U-groove weld not nail shape.                                  U shape is for samples included in Figure 8 (3 mm). Here we were referring to samples in Figure 9.
  5. line 371 First Figure 12 is redundant because the same pictures was showed below line 375                                                                            A problem with figure caption was amended.
  6. The authors of the article mention the interdendritic region, but there is no data that would clearly indicate this type of structure.                                 An explanation of how segregation ratio in interdendritic region is calculated was included.  In as-cast condition those interdendritic regions can be easily detected by local chemical analysis by EDX.
  7. The authors assume that their previous articles are known and omit some explanations. In my opinion, this should be avoided in the future.              We do not understand this comment. References to our previous articles were only included in Line 110 with extensive explanation of scope and conclusions, Line 168 to highlight that the Varestraint test bench has been previously used in other research works and in Line 421 to clarify why we used the LBW parameters included in this work. In the last case, the reference is given for those who want to check why we decided to used reported LBW parameters.
  8. In welding processes, in particular laser welding, more talks about crystallization than solidification, because in this process we have a directional grain growth.                                                                   Directional solidification grain growth is more evident in investment casting. We prefer to talk about solidification, because segregation of Nb between liquid and solid was critical to get a terminal liquid with a composition close to the eutectic.
  9. Studying welding process based on obtained trial welds, some defects analysis are recommended, are there any welding defects beside cracks, for example porosity etc. is weld build is proper...                                  LBW process parameters (spot diameter, welding speed, shielding conditions) were optimized in order to avoid pores, undercutting, oxidation,… Bead on plate samples were radiographiated. Some samples had small pores (lower than 0.7 mm in diameter) and crater cracks and the end of welds. This was not considered relevant for current study.

Round 2

Reviewer 1 Report

The paper has been modified, but I only would like to point out that the scale bars in some photographs are recommended to be modified visible for readers.

Author Response

The paper has been modified, but I only would like to point out that the scale bars in some photographs are recommended to be modified visible for readers.

Answer: Scale bars in micrographs have been edited again. A new version with new images has been submitted.
